# Laboratory-confirmed influenza infection and acute myocardial infarction among United States senior Veterans

**Yinong Young-Xu**[1]*, **Jeremy Smith**[1☯], **Salaheddin M. Mahmud**[2☯], **Robertus Van Aalst**[3,4☯], **Edward W. Thommes**[4,5‡], **Nabin Neupane**[1‡], **Jason K. H. Lee**[6,7‡], **Ayman Chit**[4,6☯]

**1** Clinical Epidemiology Program, Veterans Affairs Medical Center, White River Junction, Vermont, United States of America, **2** Department of Community Health Sciences, College of Medicine, University of Manitoba, Winnipeg, Manitoba, Canada, **3** Faculty of Medical Sciences, University of Groningen, Groningen, The Netherlands, **4** Sanofi Pasteur, Swiftwater, Pennsylvania, United States of America, **5** Department of Mathematics & Statistics, University of Guelph, Guelph, Ontario, Canada, **6** Leslie Dan School of Pharmacy, University of Toronto, Toronto, Ontario, Canada, **7** Sanofi Pasteur, Toronto, Ontario, Canada

☯ These authors contributed equally to this work.
‡ These authors also contributed equally to this work.
* Yinong.Young-Xu@va.gov

**Data Availability Statement:** We note that your Data Availability Statement reads "Data cannot be shared publicly because of ethical restrictions on confidential data. Data are available from the

## Abstract

### Background

Previous studies established an association between laboratory-confirmed influenza infection (LCI) and hospitalization for acute myocardial infarction (AMI) but not causality. We aimed to explore the underlying mechanisms by adding biological mediators to an established study design used by earlier studies.

### Methods

With data on biomarkers, we used a self-controlled case-series design to evaluate the effect of LCI on hospitalization for AMI among Veterans Health Administration (VHA) patients. We included senior Veterans (age 65 years and older) with LCI between 2010 through 2015. Patient-level data from VHA electronic medical records were used to capture laboratory results, hospitalizations, and baseline patient characteristics. We defined the "risk interval" as the first 7 days after specimen collection and the "control interval" as 1 year before and 1 year after the risk interval. More importantly, using mediation analysis, we examined the role of abnormal white blood cell (WBC) and platelet count in the relationship between LCI and AMI to explore the thrombogenic nature of this association, thus potential causality.

### Results

We identified 391 hospitalizations for AMI that occurred within +/-1 year of a positive influenza test, of which 31 (31.1 admissions/week) occurred during the risk interval and 360 (3.5/per week) during the control interval, resulting in an incidence ratio (IR) for AMI admission of 8.89 (95% confidence interval [CI]: 6.16–12.84). In stratified analyses, AMI risk was

Department of Veterans Affairs Veterans Health
Administration, Research and Development
Committee, for researchers who meet the criteria
for access to confidential data: Research and
Development Committee, VA Medical Center, 163
Veterans Drive, White River Junction, VT, 0500.
VHAWRJCEP@va.gov.

**Funding:** This work was supported by an
unrestricted research grant (CRADA#1037291-7)
from Sanofi Pasteur. The funder provided support
in the form of salaries for authors RVA, EWT, JKL,
and AC, but did not have any additional role in the
study design, data collection and analysis, decision
to publish, or preparation of the manuscript. The
specific roles of these authors are articulated in the
'author contributions' section.

**Competing interests:** YYX has received research
funding from Sanofi Pasteur, Sanofi, Pfizer,
Genentech, Janssen, VIR Biotechnology, and the
Office of Rural Health Resource Center- Eastern
Region. SMM has received research funding from
Assurex, GSK, Merck, Pfizer, Roche and Sanofi,
and is/was a member of advisory boards for GSK
and Sanofi. RVA, JKL, EWT and AC are employees
of Sanofi Pasteur. The remaining authors have
nothing to disclose. This does not alter our
adherence to PLOS ONE policies on sharing data
and materials.

significantly elevated among patients with high WBC count (IR, 12.43; 95% CI: 6.99–22.10) and high platelet count (IR, 15.89; 95% CI: 3.59–70.41).

## Conclusion

We confirmed a significant association between LCI and AMI. The risk was elevated among those with high WBC or platelet count, suggesting a potential role for inflammation and platelet activation in the underlying mechanism.

## Introduction

Young-Xu and colleagues found an estimated 10,674 emergency room visits, 2,538 hospitalizations, and 3,793 underlying respiratory or circulatory deaths among United States Veterans that were attributable to influenza each year from 2010 through 2014 [1,2]. Some studies have tried to improve the accuracy of influenza-attributable disease burden estimates [3–6], while others have worked to elucidate the causal pathways by discovering a strong association between laboratory-confirmed influenza infection (LCI) and acute myocardial infarction (AMI) [7–10]. Recently, Kwong et al. made a critical advancement in our understanding of this relationship using robust, longitudinal patient data containing LCI results and AMI diagnosis from a Canadian population [11]. Using self-controlled-case-series (SCCS) design, the authors reported an incidence ratio (IR) for AMI during the week post infection compared to the control interval of 6.05 (95% confidence interval [CI]: 3.86–9.50). These findings lend strong evidence for the causal link between LCI and AMI, thereby reinforcing guidelines [12] that advocate for influenza vaccination in persons aged 65 years or older to protect against ischemic coronary events.

Although the study by Kwong et al. has been replicated by research from Scotland and Demark [13,14], an independent replication of their study by another North American population, first time in the United States, would provide additional support to the causal relationship between LCI and AMI, especially among the elderly where influenza-attributable disease burden is the greatest [1]. Establishing consistency, though a laudable effort in establishing causality, is not enough. It is also important to examine and qualify known health risks in minority and underserved populations. To help inform better prevention and case management paradigms, we aimed to decipher the underlying mechanism by which influenza infections cause AMI as pro-inflammatory and pro-coagulant changes post infection are suspects [13]. Studies have reported a link between influenza infections and change in white blood cell counts [15,16], and change in platelet counts (PC) [17,18]. Researchers have also linked changes in WBC and in PC to myocardial infarction [19,20]. Only a few case studies, however, have put all three—influenza infection, change in WBC/PC, and myocardial infarction—together [15,16], and no study to our knowledge has quantified the mediating effects of WBC/PC. To both replicate previous epidemiological findings of the association between LCI and AMI as well as deepen our understanding of the potential biomechanism to establish causation, we examined the relationship between LCI and AMI using detailed clinical and laboratory information available via the VHA Corporate Data Warehouse (CDW), a wealth of real-world clinical data. While verifying the findings by Kwong et al. in a US Veteran population aged 65 years or older (hereinafter referred to as senior Veterans), we aim to quantify the mediating role of white blood cell count and platelet count in the relationship between LCI and AMI through stratified and mediation analyses.

## Materials and methods

We obtained ethics approval from the institutional review board at White River Junction VA Medical Center (#1037291–7). All study procedures were carried out in compliance with federal and institutional ethical guidelines. The requirement to obtain informed consent from study participants was waived as there was no more than a minimal risk to the privacy of individuals as the data were analyzed anonymously. There was no patient or public involvement in the study.

### Study design and data sources

The VHA is the largest integrated health care system in the United States (US), providing clinical care to over 9 million Veterans at more than 170 medical centers and 1,074 community-based outpatient clinics [21]. We obtained de-identified electronic medical records (EMR) data for VHA-enrollees and administrative health records from Medicare fee-for-service files. These Medicare records supplement the VHA database as many VHA patients seek health care outside the VHA system once they turn 65 and qualify for additional benefits. The VHA EMR data were available via CDW and contain detailed records on hospitalizations, outpatient visits, medications, laboratory tests and results. Each patient is assigned a unique identification number that allows for longitudinal follow-up.

We included VHA-enrolled Veterans who underwent testing for one or more respiratory viruses between July 1, 2010, and June 30, 2015, were 65 years of age or older at the time of testing, and were hospitalized for AMI between July 1, 2009, and June 30, 2016.

The respiratory specimens were collected and tested for influenza A and B and the following respiratory viruses: respiratory syncytial virus, adenovirus, coronavirus, enterovirus (including rhinovirus), parainfluenza virus, and human metapneumovirus. To avoid capturing multiple exposures for the same illness episode, we excluded positive specimens that were obtained within 14 days after a previous positive specimen from the same patient. The outcome was hospitalization with AMI as the principal discharge diagnosis [International Classification of Diseases, Ninth Revision code 410]. We restricted the analysis to the first event in a care episode by excluding transfers between hospitals and admissions within 30 days after a previous hospital discharge for AMI. There is often a lag between influenza infection, symptom onset, and subsequent laboratory testing for influenza. Therefore, we excluded AMI cases if the positive influenza specimen were obtained during the hospitalization for AMI, as we could not determine the temporal relationship between the influenza exposure and the cardiac outcome.

### Patient characteristics

Characteristics collected during the study period included demographics, comorbidities, and health care utilization. Demographics comprised age, sex, race, and priority level of VHA care. Priority level of VHA care serves as a surrogate measure for socioeconomic status because it is based on military service history, disability rating, income level, qualification for Medicaid, and other governmental benefits [22]. Veterans' residence was categorized using Rural-Urban Community Area (RUCA) code aggregations, where rural areas included RUCA codes 7.0, 7.2 to 7.4, 8.0, 8.2 to 8.4, 9.0 to 9.2, 10.0, and 10.2 to 10.6 [23].

### Statistical analysis

We adopted the same approach as that of Kwong and colleagues thus enable comparison of our results. The main statistical analysis was based on the self-controlled case-series design, as shown in Fig 1. Two event rates were calculated. One was AMI hospitalization rate during the

risk period where LCI, the exposure, occurred within 7 days prior to the AMI episode. Another is AMI hospitalization rate during the control period where there was no LCI around the time of AMI. This could be viewed as a baseline AMI hospitalization rate for a patient. The ratio of these two AMI rates (incidence ratio)—risk period AMI rate over control period AMI rate—is the measure of association between LCI and risk of AMI hospitalization.

a. The date of respiratory specimen collection served as the index date for defining the exposure (LCI) because the date of symptom onset was generally not available even though patients might have been symptomatic before their healthcare visit. The exact date of infection onset could not be determined.

b. We defined the observation period as the interval from 1 year before to 1 year after the index date, and we included in our analyses patients who had at least one admission for AMI (cases) during this period. The observation time was truncated in this manner to minimize time-varying confounding, since the self-controlled case-series design does not control for time-varying confounding.

c. In the primary analysis, we defined the "risk interval" as the first 7 days after the index date and the "control interval" as all other times during the observation period (i.e., 52 weeks before the index date and 51 weeks after the end of the risk interval) (Fig 1).

We estimated the incidence ratio as the AMI hospitalization rate during the risk interval divided by the hospitalization rate during the control interval using a fixed-effects conditional Poisson regression model [24]. In addition to the primary analysis that defined the risk interval as days 1–7 after the index date, we also considered narrower risk intervals (days 1–3 and days 4–7) and wider risk intervals (days 8–14 and days 15–28).

We stratified SCCS analyses by white blood cell count (low or elevated beyond the normal range of 4,500 to 11,000 WBCs per microliter), platelet count (low or elevated beyond the normal range of 150,000 to 400,000 per microliter), pneumonia diagnosis, record of influenza vaccination. To further examine the impact of age, we divided our study population into two groups at 75 years of age, an established risk factor for MI [25] and very close to the median age of our study population, 76. In a SCCS design, estimation is within individuals, and the effects of any time-fixed confounding factors (e.g., characteristics such as family history) cancel out. As a result, individual-specific characteristics were not included as main effects

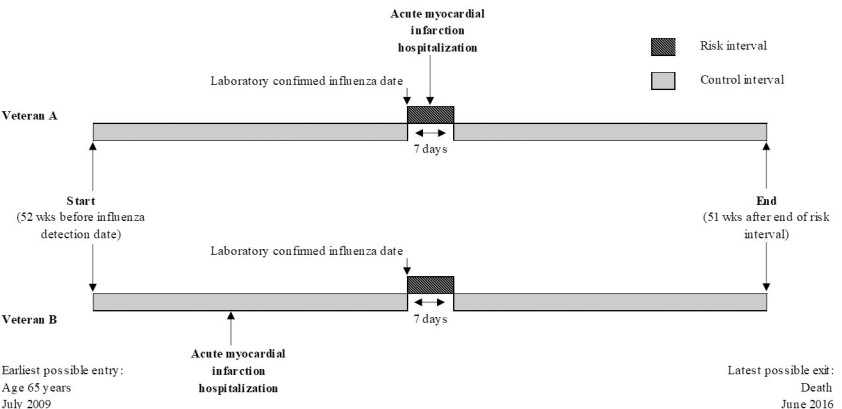

**Fig 1. Study design**\*. \* Note: Figure design is adapted from Kwong, et al. [11] Veteran A represents a person who is infected with influenza and is hospitalized for acute myocardial infarction at any time during the 7-day risk interval (dark-shaded areas) after exposure. Veteran B represents a person infected with influenza who has an acute myocardial infarction during the control interval (light-shaded areas). The study assessed the relative incidence of acute myocardial infarction during the risk interval as compared with the control interval. Note that the figure is not to scale.

(covariates) in the analysis. However, certain time-fixed covariates may act as effect modifiers. For example, the association between LCI and AMI might be age dependent. To investigate such effects, interactions between covariates, such as age group, and the exposure (LCI) were included in the analysis. For each subgroup analysis, we performed a likelihood ratio test to test the hypothesis that no interaction term should be included in the model.

We then performed mediation analysis that included a series of stepwise modeling, as proposed by Baron and Kenny in 1986 [26]. The goal of our mediation analysis was to explore potential underlying mechanisms (e.g., changes in WBC and PC) by which LCI could lead to AMI. Four steps involving three equations were required in the Baron and Kenny approach to establish mediation. We adapted these three equations to our current analysis:

Equation 1. AMI is associated with independent variable LCI

Equation 2. AMI is significantly associated with both the independent variable LCI and a mediating variable (e.g., WBC)

Equation 3. The mediating variable is significantly associated with the independent variable LCI

The mediation analysis was performed twice, with two different designs. First, we used the SCCS design and fixed-effects conditional Poisson regression model to estimate the three equations listed above. To apply the fixed-effects conditional Poisson regression model, we had to assume that WBC and PC were in the normal range whenever there was no evidence to suggest otherwise, including when they were not measured. Second, we used a survival model as additional analysis to account for the small sample size of the SCCS analysis and assumptions we made for it to work. In the survival model, we examined a classifying exposure (LCI) during the follow-up period. To control for this potential survival bias, we used a time-dependent Cox regression. As a result, LCI, WBC and PC measurements were all treated as time-dependent covariates in the Cox regression models. Follow-up began on the first day of September, which is the month when influenza testing starts to become more prevalent during each influenza period. The observation period ended on the date of disenrollment from either VHA or Medicare Part A or B, end of each influenza season, date of death, or date of an AMI hospitalization, whichever occurred first.

All statistical tests were two-tailed, and p-values of less than 0.05 denoted statistical significance. Analyses were performed with SAS software, version 9.4 (SAS Institute).

## Results and discussion

### Testing episodes and patient demographics

Among 54,096 influenza testing episodes in senior Veterans during the study period, 8,259 testing episodes (15%) were positive for LCI (Fig 2). The final data for the primary analysis consisted of 391 hospitalizations for AMI following LCI among 373 patients.

The median age was 76 years (interquartile range: 68–84), 4% of the episodes the patient was female, 8% of the episodes followed a previous hospitalization for AMI, many episodes involved patients that had established cardiovascular risk factors (57% diabetes, 75% dyslipidemia, and 94% hypertension), and patients in 89% of the episodes were vaccinated against influenza for that influenza season (Table 1). Most infections (82%) were due to influenza A.

### AMI risk following LCI

There were 31 admissions for AMI (31.1 admissions per week) during the risk interval and 360 (3.5 admissions per week) during the control interval (IR: 8.89; 95% CI: 6.16–12.84). The

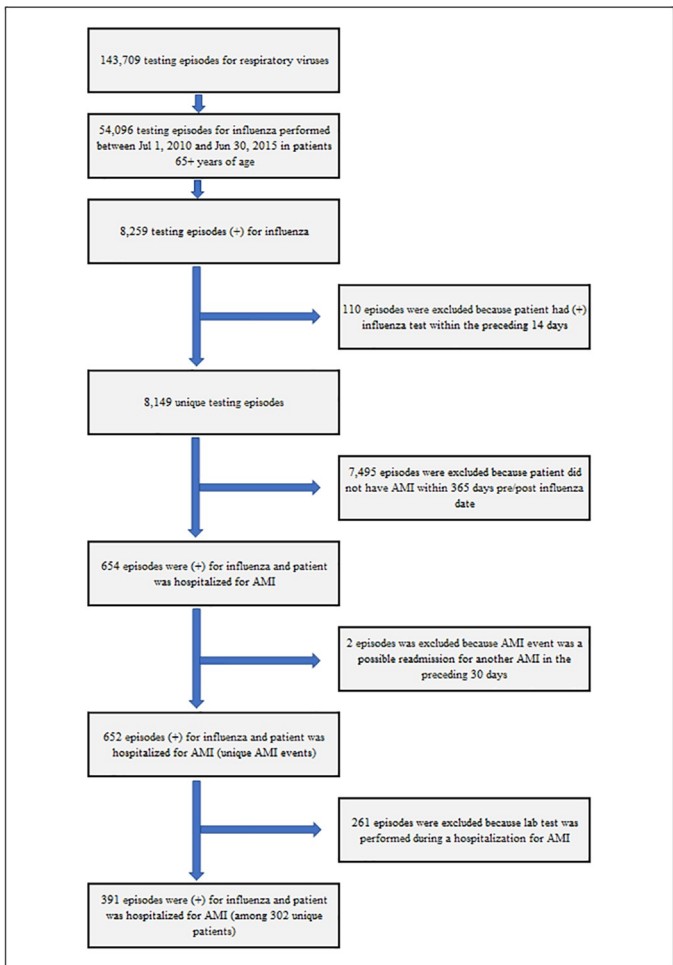

**Fig 2. Influenza testing episodes included in the study.**

IRs for days 1–3 and days 4–7 were elevated, while no significant increase in incidence was observed on days 8–14 or days 15–28. As a result, the rest of the analysis focused on days 1–7.

## Subgroup analyses

In the subgroup analyses (Table 2), a more elevated incidence ratio for AMI following LCI was observed among senior Veterans older than 75 years (IR = 11.92) compared to those aged between 65 and 75 years (IR = 5.80). The difference in IRs between the two age groups was borderline significant (P = 0.07 for interaction). Following the 391 LCI episodes, the majority (91%) of patients had biomarkers—WBC and PC—measured within the first 7 days following the sample collection for influenza testing. The IRs were higher for elevated WBC and elevated PC, but these differences were not statistically significant (P = 0.16 and 0.44 for interactions, respectively). Neither influenza vaccination nor concomitant pneumonia diagnosis impacted the IRs, with p-values for interaction at 0.61 and 0.92, respectively. The log likelihood ratio tests all favor the model without the interaction in the SCCS analysis.

In additional subgroup analyses, we explored the impact of known risk factors for cardiovascular disease (CVD). An elevated risk of acute myocardial infarction after influenza infection was observed among those without a diagnosis of dyslipidemia (IR = 11.94; 95% CI: 7.22–

**Table 1. Baseline characteristics of patients who tested positive for influenza and who had an AMI within the observation period.**

| Characteristic[a] | Total (%) |
|---|---|
| Population (episodes) | 391 (100) |
| Age at LCI, median (IQR) | 76 (69–84) |
| Age at LCI | |
| 65–74 | 179 (46) |
| 75+ | 212 (54) |
| Male | 377 (96) |
| Race | |
| black | 56 (14) |
| white | 314 (80) |
| other | 13 (3) |
| (missing) | 8 (2) |
| Rural | 115 (29) |
| VHA Priority level | |
| 1–4 | 118 (30) |
| 5–8 | 273 (70) |
| Prior AMI hospitalization[b] | 30 (8) |
| Dyslipidemia | 227 (58) |
| Diabetes | 199 (51) |
| Hypertension | 329 (84) |
| Vaccinated[c] | 347 (89) |
| Influenza type | |
| A (untyped) | 305 (78) |
| A H1N1 | 8 (2) |
| A H3 | 4 (1) |
| B | 41 (10) |
| A + B | 15 (4) |
| Unknown | 18 (5) |
| Test type[d] | |
| Antibody | 1 (0) |
| Antigen | 60 (21) |
| PCR | 218 (76) |
| PCR + Antigen | 6 (2) |
| PCR + Other | 3 (1) |
| Elevated platelet level | 15 (4) |
| Elevated WBC level | 121 (31) |
| Pneumonia dx within +/- 7 days LCI | 40 (10) |

[a] N (%) except where specified.

[b] AMI during year prior to start of study window.

[c] rec'd flu vaccine during season and at least 2 weeks prior to LCI.

[d] including only known test types.

19.73), or diabetes (IR = 14.06; 95% CI:9.09–21.73), or hypertension (IR = 17.54; 95% CI:8.65–35.56). In contrast, those with a diagnosis of dyslipidemia, diabetes, or hypertension, had lower risk of acute myocardial infarction after influenza infection: IR = 6.79, 95% CI:3.95–11.66; IR = 4.33, 95% CI:2.13–8.78; and IR = 7.40, 95% CI:4.80–11.41, respectively. Moreover, the difference in incidence ratios between the two groups was statistically significant for

**Table 2. SCCS subgroup analyses comparing incidence ratios for acute myocardial infarction after laboratory-confirmed influenza infection.**

| Subgroup (# episodes) | Incidence Ratio (CI) | P-value for Interaction Term | P-value for Log Likelihood Ratio Test |
|---|---|---|---|
| White Blood Cell Count* | | | |
| Normal | 7.13 (4.23–12.05) | | |
| Low | 8.85 (2.72–28.78) | 0.74 | |
| Elevated (95) | 12.43 (6.99–22.10) | 0.16 | 0.17 |
| Platelet Count* | | | |
| Normal | 8.66 (5.81–12.92) | | |
| Low | 8.37 (2.58–27.16) | 0.96 | |
| Elevated (13) | 15.89 (3.59–70.41) | 0.44 | 0.77 |
| Age | | | |
| < = 75 years | 5.80 (3.07–10.97) | | |
| > 75 years | 11.92 (7.59–18.72) | 0.07 | 0.06 |
| Diagnosis of Pneumonia | | | |
| No | 8.95 (6.09–13.17) | | |
| Yes (34) | 8.37 (2.58–27.16) | 0.92 | 0.92 |
| Influenza vaccination status | | | |
| No | 7.77 (4.08–14.77) | | |
| Yes | 9.56 (6.11–14.94) | 0.61 | 0.60 |
| Dyslipidemia | | | |
| Yes | 6.79 (3.95–11.66) | | |
| No | 11.94 (7.22–19.73) | 0.13 | 0.13 |
| Diabetes | | | |
| Yes | 4.33 (2.13–8.78) | | |
| No | 14.06 (9.09–21.73) | 0.005 | 0.003 |
| Hypertension | | | |
| Yes | 7.40 (4.80–11.41) | | |
| No | 17.54 (8.65–35.56) | 0.04 | 0.05 |
| Prior AMI | | | |
| Yes | 3.56 (0.49–26.15) | | |
| No | 9.36 (6.44–13.60) | 0.35 | 0.28 |
| Overall | | | |
| Senior Veterans | 8.89 (6.16–12.84) | | |
| Senior Ontario Patients [11] | 7.31 (4.53–11.79) | | |

*All measured within 1 to 7 days after influenza specimen was drawn (i.e. index date)

**Table 3. Mediation analysis using SCCS design.**

| | Model 1 | Model 2 | Model 3 | Model 4 |
|---|---|---|---|---|
| | LCI | LCI plus WBC | LCI plus PC | LCI plus WBC plus PC |
| LCI | 8.89 (6.16–12.84) | 7.13 (4.23–12.05) | 8.66 (5.81–12.92) | 7.15 (4.24–12.07) |
| WBC (low) | | 1.24 (0.34–4.50) | | 0.97 (0.28–3.35) |
| WBC (high) | | 1.74 (0.80–3.79) | | 1.74 (0.80–3.79) |
| PC (low) | | | 0.97 (0.27–3.35) | 1.24 (0.34–4.50) |
| PC (high) | | | 1.83 (0.39–8.57) | 1.83 (0.39–8.57) |
| Drop in LCI estimate | | 20% | 3% | 20% |
| Log Likelihood Ratio Test | | 0.38 | 0.77 | 0.43 |

diabetes and hypertension, with p-value 0.005 and 0.04 for interaction term, respectively. The risk of acute myocardial infarction was also elevated among those with prior hospitalization for AMI before the study period—IR = 9.36, 95% CI: 6.44–13.60 –vs those without: IR = 3.56, 95% CI: 0.49–26.15, but the interaction term was not statistically significant (p = 0.35) (Table 3).

## Mediation analysis

Our overall approach to mediation analysis is to compare the results of multivariable analysis to the main result from the univariate analysis. In the mediation analysis using SCCS design (Table 3), the original, univariable, model (model 1) involving LCI only showed an IR of 8.89, the study's main finding. In model 2, we added two binary variables into the model—low or elevated WBC—to see the impact on the main result. The IR for LCI, low WBC and elevated WBC were 7.13, 1.24 and 1.74, respectively. In other words, the main result, the IR for LCI, dropped 20% compared to the IR for LCI in model 1 (7.13 vs. 8.89). In model 3, we inserted two binary variables to model 1 –low or elevated PC—to examine the impact of PC on the main result. The IR for LCI, low PC and elevated PC were 8.66, 0.97, and 1.83, respectively. This time, the IR for LCI dropped only 3% compared to the IR for LCI in model 1 (8.66 vs. 8.89). Finally, in model 4, we inserted all four binary variables, two each for WBC and PC— low or elevated. The IR for LCI, low WBC, elevated WBC, low PC, and elevated PC were 7.15, 0.97, 1.74, 1.24, and 1.83, respectively. The IR for LCI dropped 20% compared to the IR for LCI in model 1 (7.15 vs. 8.89).

## Secondary analysis

Because these findings are dependent upon sample size and study design (e.g. SCCS), we performed an additional mediation analysis (Table 4) using survival analysis (time to event model) which does not rely on self-controls to adjust for potential confounders. As a result, we could control for measured confounders such as demographic characteristics and underlying medical conditions by including them specifically in the model and thus enabling us to quantify their associated risks. We again followed the three equations that we adapted from Baron and Kenny [18]. We first found a significant relationship of the independent variable, LCI, to the dependent variable, AMI, that was expected in Equation 1. The unadjusted hazard ratio (HR, i.e., risk for AMI) was 56.05 (95% CI: 44.02–71.36). We confirmed the second equation that implies the mediating variable (e.g. WBC) must be significantly related to the dependent variable when both the independent variable and mediating variable are predictors of the dependent variable. We then made adjustment for demographic and comorbidities specifically (Table 4, model 2). After adjusting for these characteristics, we inserted variables for WBC and PC as we did in our previous mediation analysis. The results indicated a mediating effect: HR for LCI dropped from 56.02 (95% CI: 44.02–71.36) in univariate model 1 to 4.42 (95% CI: 3.45–5.66) in model 3. In other words, the coefficient relating the independent variable to the dependent variable was much larger in absolute value (56.02 in this case) than the coefficient relating the independent variable to the dependent variable (4.42) in the regression model with both the independent variable (LCI) and the mediating variables (WBC and PC) predicting the dependent variable.

Secondary analysis using survival model included more than three and half million person-seasons of data and confirmed those risk factors that we examined in the SCCS design but did not achieve statistical significance due to small sample size, for example, older than 75. We were also able to examine other variables such as sex (male), race (white), rurality, and VHA priority. In our study population, being male was associated with a greater risk of AMI with an

**Table 4. Mediation analysis using survival model.**

| Model | model_1 | model_2 | model_3 |
|---|---|---|---|
| | **FLU** | **FLU + DEMOG + COMORB** | **FLU + DEMOG + COMORB + WBC + PLT + PNEUM** |
| **Variables** | | | |
| **LCI** | 56.05 *** (44.02–71.36) | 42.48 *** (33.36–54.11) | 4.42 *** (3.45–5.66) |
| **Inflammatory Markers** | | | |
| WBC Elevated | | | 3.38 *** (3.20–3.56) |
| WBC Low L | | | 0.62 *** (0.54–0.72) |
| WBC Unmeasured | | | 0.59 ** (0.41–0.86) |
| PC Elevated | | | 0.97 (0.85–1.10) |
| PC Low | | | 1.13 *** (1.06–1.21) |
| PC Unmeasured | | | 0.31 *** (0.21–0.45) |
| **Pneumonia** | | | |
| Pneumonia | | | 3.82 *** (3.38–4.32) |
| **Demographics** | | | |
| Rurality | | 1.05 *** (1.03–1.07) | 1.06 *** (1.03–1.08) |
| Priority 1 | | 0.89 *** (0.87–0.91) | 0.86 *** (0.84–0.88) |
| Male | | 1.36 *** (1.25–1.48) | 1.33 *** (1.22–1.45) |
| Age 75–84 | | 1.22 *** (1.19–1.25) | 1.30 *** (1.27–1.34) |
| Age 85 or older | | 1.67 *** (1.62–1.72) | 1.84 *** (1.79–1.90) |
| White | | 0.95 *** (0.93–0.97) | 0.94 *** (0.91–0.96) |
| **Comorbidities** | | | |
| Diabetes w/o Complications | | 1.19 *** (1.17–1.22) | 1.17 *** (1.14–1.20) |
| Renal Disease | | 1.36 *** (1.33–1.39) | 1.30 *** (1.27–1.34) |
| Diabetes w/ Complications | | 1.20 *** (1.16–1.23) | 1.17 *** (1.14–1.21) |
| Dementia | | 1.01 (0.97–1.05) | 1.02 (0.98–1.06) |
| Cerebrovascular Disease | | 1.20 *** (1.17–1.23) | 1.20 *** (1.17–1.23) |
| COPD | | 1.14 *** (1.11–1.17) | 1.09 *** (1.06–1.11) |
| Congestive Heart Failure | | 1.60 *** (1.56–1.64) | 1.54 *** (1.51–1.58) |
| Metastatic Cancer | | 1.42 *** (1.33–1.51) | 1.26 *** (1.19–1.35) |
| Cancer | | 1 (0.98–1.03) | 0.95 *** (0.93–0.98) |
| Rheumatoid Arthritis | | 1.05 (1.00–1.11) | 1.03 (0.97–1.08) |
| Peripheral Vascular Disease | | 1.27 *** (1.24–1.30) | 1.27 *** (1.24–1.30) |
| Peptic Ulcer Disease | | 1.05 * (1.00–1.10) | 1.04 (0.99–1.09) |
| Paraplegia/Hemiplegia | | 0.93 * (0.88–0.99) | 0.91 ** (0.85–0.96) |
| Myocardial Infarction | | 5.19 *** (5.07–5.31) | 5.21 *** (5.09–5.33) |
| Severe Liver Disease | | 1.20 ** (1.08–1.35) | 1.13 * (1.01–1.27) |
| Mild Liver Disease | | 1.05 (1.00–1.10) | 1 (0.96–1.05) |
| HIV/AIDS | | 1.40 ** (1.13–1.75) | 1.2 (0.96–1.49) |
| **Log Likelihood Ratio Tests** | | | |
| LLR test | | <0.001 | <0.001 |

Note:

*< 0.05,

**<0.01,

***<0.001

HR of 1.33 (95% CI, 1.22–1.45). Being White and having good access to healthcare (VHA priority 1) were associated with lower risk of AMI: HR 0.94 (95% CI, 0.91–0.96) and HR 0.86 (95% CI, 0.84–0.88), respectively. Living in a rural area where access to healthcare could be problematic was associated with a slight increase in the risk of AMI with an HR of 1.06 (95% CI, 1.03–1.08). Finally, we controlled for a list of comorbid conditions in the survival analysis with many of them known factors associated with elevated risk of MI. For example, patients with diabetes, even those without complications, had a HR of 1.17 (95% CI, 1.14–1.20). As the survival analysis included patients without LCI and patients who never experienced an AMI, prior AMI appeared as a strong risk factor with a HR of 5.21 (95% CI, 5.09–5.33) (Table 4).

## Discussion

Among senior Veterans, we found that the incidence of admissions for AMI was 9 times as high during the seven days after LCI compared to that during the control interval (31.1 admissions per week vs. 3.5 admissions per week). The IR point estimates were highest for those older than 75 years, and for patients with elevated WBC and PC. However, the SCCS analyses were insufficiently powered to identify differences within these subgroups. Through our secondary analysis using a survival model, we were able to explore the role that LCI has in precipitating AMI with a larger sample size. Specifically, we explored the roles of elevated WBC and PC in mediating this link.

Our findings were consistent with those reported in previous studies. The study by Kwong et al. included patients younger than 65 but analyzed patients older than 65 in a subgroup analysis. They found an IR of 7.31 (95% CI, 4.53–11.79), like the IR of 8.89 that we reported here. The magnitude of the IR in our study may have been greater because the risk for a predominantly male (96% vs. their 52%) and elderly population was higher: 75% of our study population had dyslipidemia, 57% had diabetes, and 94% had hypertension. These estimates were 38%, 49%, and 85%, respectively in the Kwong et al. study population. Additionally, we examined those with and without an influenza vaccination record. If patients with a vaccination record have influenza infection of sufficient severity to warrant testing, and therefore inclusion in our study population, their AMI risk increased to a level similar to unvaccinated patients: IR of 9.6 for vaccinated vs. IR of 7.8 for those without a vaccination record, with a p-value of 0.61 for the interaction term, implying no statistical difference (Table 2). Although our results align with Kwong et al. in this respect, neither study was designed to evaluate the effectiveness of influenza vaccines since additional data such as immune response of the vaccinated are needed. Moreover, vaccination effectiveness varied greatly, between 19% to 60%, in preventing LCI during our study period [27]. Future studies should evaluate the impact of pharmaceutical intervention, including vaccination and medications such as anticoagulants.

As well established and meticulously documented by the Framingham Heart Study, age is a risk factor for AMI. Being 75 or older contributes one full point to the Framingham risk score. Kwong et al found an IR of 7.31 (95% CI, 4.53–11.79) for those older than 65 in their study vs. those who were 65 or younger 2.38 (95% CI, 0.59–9.66). Although the p-value for interaction was marginal– 0.14 –it probably had more to do with the small sample size for the 65 or younger group, a mere 26% of their patients. In the recent Denmark study, Ohland et al found an IR of 5.1 (95% CI, 0.7–36.6) for those younger than 65, and an IR of 21.1 (95% CI, 9.8–45.6) for those 65 or older. Because slightly over half of our patients were older than 75 (54%), we investigated the impact of age on the risk of AMI using 75 as a cutoff point. We found similar results as other studies, older age is associated with a higher incidence ratio of AMI (IR = 11.92), compared to those aged between 65 and 75 years (IR = 5.80), although it

was only borderline statistically significant in the SCCS design (p = 0.07, Table 2). The different design of survival analysis (Table 4) confirmed its statistical significance (p<0.001). When age was analyzed as a continuous variable, the associated HR was 1.026 (95% CI: 1.024, 1.027) for model 2 and 1.032 (95% CI: 1.030, 1.033) for model 3 (Table 4).

An elevated incidence ratio of acute myocardial infarction after influenza infection was observed among those without a diagnosis of dyslipidemia, diabetes, or hypertension, or without prior hospitalization for acute myocardial infarction before the study period, suggesting LCI might raise more the risk of AMI among those without preconditions. The last one was consistent with findings from Kwong and colleagues. As we found an IR of 9.36 (95% CI, 6.44–13.60) for those without prior hospitalization for AMI and an IR of 3.56 (95% CI, 0.49–26.15) for those with prior hospitalization for AMI, they found an IR of 6.93 (95% CI, 4.24–11.33) and an IR of 3.53 (95% CI, 1.12–11.14), correspondingly. Both interaction terms were not statistically significant, p = 0.35 in our analysis (Table 3) and p = 0.29 in theirs [11] Unfortunately, they did not perform similar analysis by risk factors for CVD. Further analysis regarding variant confounders such as medication, diet and exercise might be useful to elucidate the lower risk of AMI associated with LCI among patients with diagnosed CVD risk factors. Perhaps medical management brought down their risk for AMI to a level even below that of those who have risk factors, although SCCS design is not able to adjust for time-varying confounding variables that could change between the risk and the control period.

Atherosclerotic-plaque disruption and superimposed thrombosis have been proposed as underlying biological mechanisms for the association between LCI and AMI [28]. The acceptable WBC range in adults is generally between 3,500 and 10,500 white blood cells per microliter of blood. It could fluctuate beyond the normal range, mostly elevated, depending on the patient, comorbidity, and time from the onset of influenza infection [29,30]. In both children and adults, infections are the most common cause of an elevated platelet count [31]. Following episodes of LCI, we saw 30% with elevated WBC but only 4% with elevated PC; 10% with low WBC and 10% with low PC. The rest were normal: 59% for WBC, and 85% for PC. Through the processes discussed above, influenza infection creates a thrombogenic environment through platelet activation and endothelial dysfunction, as we see evidence of elevated WBC and PC. More detailed measurements are needed to carefully ascertain the sequence of events. Nevertheless, our data lend support to the current understanding of biological mechanisms underlying the link between LCI and AMI. However, it also points to deficiencies in our knowledge, as the combined effect of WBC and PC explained only 20% of the association between LCI and AMI in the SCCS design (IR of 8.89 in model 1 drops to 7.15 in model 4, Table 3). Additionally, we saw both a post-infection dip and rise in these counts, perhaps reflecting acute-infarction phase. In future studies, we hope to elucidate the temporal relationship between these measures and the timing of both LCI and AMI, and types of WBC.

When these biomarkers were not measured for a patient, they were labeled as unknown WBC and PC. And these were shown to be associated with a lower risk of developing AMI: 0.59 (95% CI: 0.41–0.86) and 0.31 (95% CI: 0.21–0.45), respectively (Table 4, survival analysis). However, the comparison group consists of patients with normal WBC and PC. It is possible that a lack of testing might signal better health than those with laboratory values within the normal ranges. This was different for the mediation analysis conducted with SCCS design, which did not have a category for unknown level of WBC and PC. Since 91% of the episodes had these biomarkers tested during the risk interval, we designated the remaining 9% as normal, as well as their levels during the control intervals.

## Limitations

There are important limitations to our study. A lack of more precise data prevented us from being able to study the temporal relationship between bio-measures and the timing of LCI and AMI, the types of WBC, and the nature of AMI. This lack could also have contributed to our finding that the combined effect of WBC and PC explained only 20% of the association between LCI and AMI, although ours is the first study to quantify the mediating effect of WBC/PC on the link between influenza infection and acute myocardial infarction. Better quality and larger quantity of data would enable more sophisticated mediation analysis in the future. We used the specimen collection date as the index date, but infection might have occurred prior to specimen collection. To account for this, we conducted a sensitivity analysis that incorporated induction intervals of 2, 4, and 7 days, and confirmed that the specimen collection date was a reasonable approximation for the index date. Unfortunately, not all influenza test types were recorded, but we were able to extract close to three quarters of them (288 out 391). Our study comprised elderly (65 or older) and mostly male (96%); as a result, our findings are not widely generalizable.

There is the possibility for residual confounding in our study. In the survival analysis, we observed a HR of 56.05 (Table 4, model 1) that is much higher than the IR of 8.89 observed in the SCCS analysis, a design that is much less susceptible to confounding. We identified a HR of 4.42 after controlling for potential confounders (Table 4, model 3). A SCCS involving Veterans showed a remarkable increase in AMI risk during the first 15 days after hospitalization for acute bacterial pneumonia, to a risk that was 48 times higher than any 15-day period during the year before or after the onset of infection [32]. In a study from Denmark, researchers found that, "following onset of an S. pneumoniae infection, the acute myocardial infarction IR was 37.1 and for stroke the IR was 43.3 during the 1–3 days risk period among 40–64 year olds" [14]. These large differences between estimates from various models suggest the potential for reducing residual confounding with a combination of precise data, effective study design, and appropriate statistical modeling.

## Conclusions

Scientific advance is propelled by originality and new findings but is also supported by repeated verification, often in understudied populations, and by groundwork that could lay down the foundation for future endeavors. In this study, we confirmed a significant association between LCI and AMI among a uniquely vulnerable population—senior Veterans, for the first time in the United States. The risk was elevated among patients with elevated WBC or PC, confirming a role for inflammation and platelet activation in the underlying mechanism that could pave the way for pharmaceutical interventions.

## Author Contributions

**Conceptualization:** Yinong Young-Xu, Salaheddin M. Mahmud, Robertus Van Aalst, Edward W. Thommes, Ayman Chit.

**Data curation:** Yinong Young-Xu, Jeremy Smith, Nabin Neupane.

**Formal analysis:** Yinong Young-Xu, Jeremy Smith, Nabin Neupane.

**Funding acquisition:** Yinong Young-Xu, Ayman Chit.

**Investigation:** Yinong Young-Xu, Jeremy Smith, Robertus Van Aalst, Edward W. Thommes, Nabin Neupane, Jason K. H. Lee, Ayman Chit.

**Methodology:** Yinong Young-Xu, Jeremy Smith, Salaheddin M. Mahmud, Robertus Van Aalst, Edward W. Thommes, Jason K. H. Lee, Ayman Chit.

**Project administration:** Yinong Young-Xu.

**Resources:** Yinong Young-Xu.

**Supervision:** Yinong Young-Xu.

**Validation:** Yinong Young-Xu.

**Writing – original draft:** Yinong Young-Xu, Salaheddin M. Mahmud, Robertus Van Aalst, Edward W. Thommes, Jason K. H. Lee, Ayman Chit.

**Writing – review & editing:** Yinong Young-Xu, Salaheddin M. Mahmud, Robertus Van Aalst, Edward W. Thommes, Nabin Neupane, Jason K. H. Lee, Ayman Chit.

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
