## [Decision Letter · Decision Letter 0]

2 Sep 2020

PONE-D-20-22385

Laboratory-Confirmed Influenza Infection and Acute Myocardial Infarction among United States Senior Veterans

PLOS ONE

Dear Dr. Young-Xu,

Thank you for submitting your manuscript to PLOS ONE. After careful consideration, we feel that it has merit but does not fully meet PLOS ONE’s publication criteria as it currently stands. Therefore, we invite you to submit a revised version of the manuscript that addresses the points raised during the review process.

Please justify the rationale for conducting this study, as the concern raised by the reviewer and explain what this study adds further to the existing knowledge.

We look forward to receiving your revised manuscript.

Kind regards,

Muhammad Aziz Rahman, MBBS, MPH, CertGTC, GCHECTL, PhD

Academic Editor

PLOS ONE

Journal Requirements:

2. In your Methods section please include the dates upon which authors accessed the clinical data sources used in this study.

"YYX has received research funding from Sanofi Pasteur, Sanofi, Pfizer, Genentech, Janssen, VIR Biotechnology, and the Office of Rural Health Resource Center-Eastern Region. SMM has received research funding from Assurex, GSK, Merck, Pfizer, Roche and Sanofi, and is/was a member of advisory boards for GSK and Sanofi. RVA, JKL, EWT and AC are employees of Sanofi Pasteur. The remaining authors have nothing to disclose."

We note that one or more of the authors have an affiliation to the commercial funders of this research study : Sanofi Pasteur.

3.1. Please provide an amended Funding Statement declaring this commercial affiliation, as well as a statement regarding the Role of Funders in your study. If the funding organization did not play a role in the study design, data collection and analysis, decision to publish, or preparation of the manuscript and only provided financial support in the form of authors' salaries and/or research materials, please review your statements relating to the author contributions, and ensure you have specifically and accurately indicated the role(s) that these authors had in your study. You can update author roles in the Author Contributions section of the online submission form.

3.2. Please also provide an updated Competing Interests Statement declaring this commercial affiliation along with any other relevant declarations relating to employment, consultancy, patents, products in development, or marketed products, etc.  

5. Please upload a new copy of Figure 2 as the detail is not clear. Please follow the link for more information: https://blogs.plos.org/plos/2019/06/looking-good-tips-for-creating-your-plos-figures-graphics/

Reviewers' comments:

Reviewer's Responses to Questions

**Comments to the Author**

1. Is the manuscript technically sound, and do the data support the conclusions?

Reviewer #1: Partly

Reviewer #2: Yes

2. Has the statistical analysis been performed appropriately and rigorously? 

Reviewer #1: No

Reviewer #2: Yes

3. Have the authors made all data underlying the findings in their manuscript fully available?

Reviewer #1: No

Reviewer #2: Yes

4. Is the manuscript presented in an intelligible fashion and written in standard English?

Reviewer #1: Yes

Reviewer #2: Yes

5. Review Comments to the Author

Reviewer #1: It is a well-written article, although it was adapted from a previous study with similar design. It is somewhat lacking in originality and appealing new findings. I believe further analysis and edits and could improve the quality of this paper.

1. In line number 135-137 and 143, the authors described Laboratory confirmed influenza as the exposure variable and acute MI hospitalization as the outcome variable for the purpose of this study. The authors concluded that they found “significant association between LCI and AMI among senior Veterans”. However, the have compared with the “control interval” which is 52 weeks before the index date (Influenza test date) and 51 weeks after the end of the risk interval (the first 7 days after the index date). In general, exposure should be prior to any outcome for measurement of an association between exposure and outcome. How AMI episode before the influenza test can be included for association analysis? I think there is proper explanation and authors could consider including this clearly in the methods section.

2. More than half of the patients had known high risk factors for CVD (table 1). It would be good to see if patients with the similar co morbidity were stratified and compared for AMI after LCI.

3. Fig 1 showing risk interval as “Dark shaded area” which conflicts with the description of this figure in line no 392-394.

4. This study included 30 episodes of previous AMI hospitalization during year prior to start of study window (Table 1). Excluding old MI episodes from the study could provide more insight.

5. Out of 391 influenza tests, we see a total 288 test by test type in Table 1. Where is other tests?

6. Among those tests, one is antibody test. Which antibody? Is it reliable to indicate an acute influenza infection?

7. Throughout the analysis, authors compared rural (115) with non-rural patients. It would be great if authors define rural in their description of the study design.

8. As the 96% episodes included in this study are “male’, gender is not a strong factor to include in the comparison (line 234, 306 and table 4).

9. Overall, it would be an added value if authors clearly describe what new knowledge were added by this manuscript.

Reviewer #2: 1. Please choose one heading for the introduction section in line 77, either introduction or background.

2. As this study is also exploring the mediating role of WBC and PC, I would suggest the inclusion of previous knowledge on WBC and PC count in relation to influenza and acute myocardial infarction in the introduction section and also in the discussion section.

3. Line 184-185: Is there any justification why age 75 years was taken as a cut-off point to dichotomise the age variable? Why did not the authors consider age as a continuous variable?

4. The aim of mediation analysis using a survival model (Table 4) is to adjust for potential variables. However, I would recommend elaborating the description of table 4 in the result section - what significant variables (including comorbidities) imply.

5. As this study has tried to examine the impact of age, please discuss what other studies have found in the discussion section as well.

6. Line 311 – 312: Please include the p-value here to make it clearer that there was no significant difference between the IR between those having vaccination record and without a vaccination record.

7. Line 333: 8.89 in model 1, not model 4. Please correct it.

8. In conclusion, line 373-374, the risk to the patient of older age might be misleading as no statistical significance was observed.

6. PLOS authors have the option to publish the peer review history of their article (what does this mean?). If published, this will include your full peer review and any attached files.

Reviewer #1: No

Reviewer #2: No

---

## [Author Response · Author response to Decision Letter 0]

27 Oct 2020

We have included all our responses in a "response to reviewers" letter.

---

## [Decision Letter · Decision Letter 1]

18 Nov 2020

Laboratory-Confirmed Influenza Infection and Acute Myocardial Infarction among United States Senior Veterans

PONE-D-20-22385R1

Dear Dr. Young-Xu,

We’re pleased to inform you that your manuscript has been judged scientifically suitable for publication and will be formally accepted for publication once it meets all outstanding technical requirements.

Kind regards,

Associate Professor Dr Muhammad Aziz Rahman,

MBBS, MPH, CertGTC, GCHECTL, PhD

Academic Editor

PLOS ONE

---

## [Editor Report · Acceptance letter]

2 Dec 2020

PONE-D-20-22385R1 

Laboratory-Confirmed Influenza Infection and Acute Myocardial Infarction among United States Senior Veterans 

Dear Dr. Young-Xu:

I'm pleased to inform you that your manuscript has been deemed suitable for publication in PLOS ONE. Congratulations! Your manuscript is now with our production department. 

Kind regards, 

on behalf of

Associate Professor Dr. Muhammad Aziz Rahman 

Academic Editor

PLOS ONE